

# Recombination of endophytic bacteria in asexual plant *Ligusticum chuanxiong* Hort. caused by transplanting

Wanting Xiao[1,2], Zhanling Zhang[1,2], Hai Wang[1,3], Guiqi Han[1,2,3], Zhu-Yun Yan[1,2] and Dongmei He[1,2]

[1] Key Laboratory of Characteristic Chinese Medicinal Resources in Southwest, Chengdu, Sichuan, China
[2] School of Pharmacy, Chengdu University of Traditional Chinese Medicine, Chengdu, Sichuan, China
[3] College of Medical Technology, Chengdu University of Traditional Chinese Medicine, Chengdu, Sichuan, China

Corresponding author
Dongmei He, hedongmei@cdutcm.edu.cn

## ABSTRACT

**Background**. Long-term asexual reproduction can easily lead to the degradation of plant germplasm, serious diseases and insect pests, reduction of production and even catastrophic crop failure. "Mountain Breeding and Dam Cultivation" is the main cultivation mode of *Ligusticum chuanxiong* Hort., which successfully avoided the germplasm degradation caused by long-term asexual reproduction. The recombination of endophytic fungi of *L. chuanxiong* caused by off-site transplantation was considered to be an important reason for its germplasm rejuvenation. However, whether bacteria have the same regularity is not yet known.

**Methods**. In this study, we carried out the experiment of cultivating propagation materials of *L. chuanxiong* in different regions and transplanting them to the same region. High-throughput sequencing was performed to analyze the bacterial communities in *L. chuanxiong* and its soil.

**Results**. The results showed that after transplanting, the plant height, tiller number, fresh weight, *etc.* of *L. chuanxiong* in mountainous areas were significantly higher than those in dam areas. At the same time, significant changes had taken place in the endophytic bacteria in reproductive material stem nodes (Lingzi, abbreviated as LZ). The diversity and abundance of bacteria in dam area LZ (YL) are significantly higher than those in mountainous area LZ (ML). The relative abundance of bacteria such as Xanthobacteraceae, Micromonosporaceae, Beijerinkiaceae, Rhodanobacteria, in ML is significantly higher than YL, mainly classified in Proteobateria and Actinobacteriota. In addition, the abundance advantage of Actinobacteriota still exists in MY (underground mature rhizomes obtained by ML). Meanwhile, the bacterial community was different in different area of transplanting. The diversity of bacterial communities in dam soil (YLS) is significantly higher than that in mountain soil (MLS). MLS had more Acidobacteriota than YLS. Comparative analysis showed that 74.38% of bacteria in ML are found in MLS, and 87.91% of bacteria in YL are found in YLS.

**Conclusions**. We can conclude that the community structure of endophytic bacteria recombined after the transplantation of *L. chuanxiong*, which was related to the bacterial community in soils. Moreover, after transplanting in mountainous areas, LZ accumulated more potentially beneficial Actinobacteriota, which may be an important reason for promoting the rejuvenation of germplasm in *L. chuanxiong*. However, this hypothesis requires more specific experiments to verify. This study provided a new idea

that off-site transplanting may be a new strategy to restore vegetative plant germplasm resources.

## INTRODUCTION

Germplasm degradation is a common problem in asexual reproduction (*Thomas-Sharma et al., 2016*). Although asexual reproduction maintains the continuity of species through clonal ramets in space (*Lei, 2010*), the genetic homogeneity caused by long-term asexual reproduction will increase the probability of disease and pest infection more than sexual reproduction (*Yang & Kim, 2016*). The problem of germplasm degradation often occurs in the asexual propagation of crops such as sweetpotato (*Gibson & Kreuze, 2015*), banana (*Jacobsen et al., 2019*) and garlic (*Khar et al., 2020*). At the same time, germplasm degradation will also occur during the cultivation of vegetatively propagated medicinal plants such as *Gastrodia elata* (*Jiang et al., 2022*), *Crocus sativus* (*Mahpara et al., 2018*), and *Bletilla striata* (*Wei et al., 2018*), resulting in low yield and poor quality of medicinal materials. The adverse effects of asexual reproduction can be improved by restoring sexual reproduction to increase genetic variability or by using tissue culture technology to obtain virus-free seedlings, which has been proved to be effective in the cultivation of potato (*Kim et al., 2020*), *Angelica sinensis* (*Butola et al., 2010*), *Pinellia ternata* (*Peng et al., 2007*), *etc.* However, these methods are weak for plants lacking sexual reproduction and difficult to achieve tissue culture in the short term.

Endophytes are microorganisms that live in healthy plant tissues without causing host plants to show obvious infection symptoms (*Allen et al., 2007*; *Sturz, Christie & Nowak, 2000*). *Amine, Paloma & Stéphane (2018)* put forward the concept of "holobiont", considering that plants and microorganisms interact and evolve each other. Some plant endophytes play a positive role in plant growth, such as increasing plant nutrition access, promoting plant hormone production, and protecting plants from pathogens (*Stéphane et al., 2005*; *Compant, Clément & Sessitsch, 2010*). The population density of endophytes mainly depends on microbial species, host genotype, host development stage and environmental conditions (*Zhiyuan, Thomas & Barbara, 2003*).

*Ligusticum chuanxiong* Hort. is a plant of *Ligusticum* in Umbelliferae. There have many useful compounds such as ligustilide, chuanxiongzine, ferulic acid and other ingredients in its rhizome, which have antioxidant, anti-inflammatory, antibacterial and other effects (*Xia et al., 2011*). In clinic, it is often used to treat cardiovascular and cerebrovascular diseases such as atherosclerosis, angina pectoris, hypertension (*Or et al., 2010*; *Haigang et al., 2018*). In addition, it is widely used in China, Japan and other countries. *L. chuanxiong* is major produced in the ecotone between mountains and dam area on the western edge of Sichuan Basin, China. As a result of long-term domestication and cultivation, there are no wild species and no flowers or seeds (*Chen et al., 2018a*). *L. chuanxiong* is cultivated by the

asexual reproduction method of "Mountain Breeding, Dam Cultivation". From January to February every year, the rhizome of some *L. chuanxiong* (Puxiong, abbreviated as PX) in the dam area (500–600 m altitude) are dug out and transplanted to the mountain area (1,000 m altitude) for reproduction. The other plants are kept to continue growing untill May, then the rhizome is harvested as medicinal part (Chuanxiong, abbreviated as CX). After "Mountain Breeding", the stem nodes (Lingzi, abbreviated as LZ) were harvested as breeding materials at the end of July of the following year. Then the LZ is planted in the dam area for circular cultivation (Fig. 1). "Mountain Breeding" is an important guarantee for the production quality and provenance of *L. chuanxiong*. In recent years, a new mode called "Dam Breeding" has appeared. However, germplasm degradation after 3-5 times circular cultivation occurred for *L. chuanxiong* obtained in this way, with serious diseases and insect pests (*Chen et al., 2018b*). Therefore, it must be transplanted back to mountainous area to restore germplasm. In our previous research, we found that the process of mountain breeding led to the recombination of endophytic fungi in LZ, with more disease resistant fungi and growth promoting fungi in the LZ, while the pathogenic fungi were reduced in it. These recombinated fungi were proveded could promote the growth and development of host and resist pathogenic bacteria (*Kang et al., 2021*). This study focused on the endophytic bacterial community structure in the process of propagation and transplantation of *L. chuanxiong*, hoping to clarify the microecological reorganization mechanism caused by the transplantation process and its contribution to the rejuvenation of asexual plant germplasm.

## MATERIALS & METHODS

### Transplantation experiment design

We conducted cultivation experiments on mountain breeding and dam breeding (Fig. 2). Two experimental sample area, Shuimo Mountain and Shiyang Dam, were selected to represent traditional "Mountain Breeding" and emerging "Dam Breeding" respectively. In February of the first year, the immature rhizomes (PX) of *L. chuanxiong* were collected from Shiyang dam area and cultivated in Shuimo Mountain area (1,073 m) and Shiyang dam area (592 m). At the end of July, the stem nodes (LZ) of the two places were collected and transplanted to Shiyang Dam for cultivation. Finally, the rhizomes were collected in May of the next year. In the whole experiment, we collected the immature rhizome of *L. chuanxiong* (PX), the whole plant (SM), LZ (ML) and its bulk soil (MLS) in mountain area, the whole plant (SY), LZ (YL) and its bulk soil (YLS) in dam area, the rhizome obtained from ML transplantation (MY) and its rhizosphere soil (MYS), the rhizome obtained from YL transplantation (YY) and its rhizosphere soil (YYS). The biological duplication of PX are three, while the biological duplication of other samples are six.

### Test materials and pretreatment

(1) Soil sample

The bulk soil was collected before planting LZ, at a depth of 10 cm underground. Rhizosphere soil was collected by root shaking method. Impurities were removed, placed
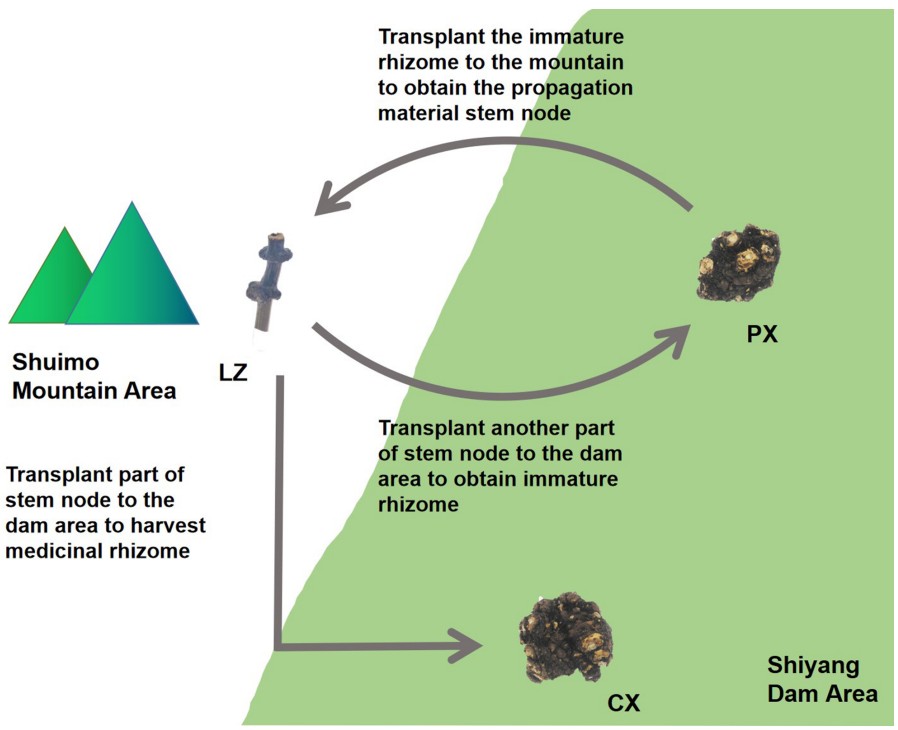

**Figure 1** **Traditional cultivation pattern of *Ligusticum chuanxiong*.** PX, immature rhizome of *L. chuanxiong*; LZ, reproductive material stem node of *L. chuanxiong*; CX, mature rhizome of *L. chuanxiong*.

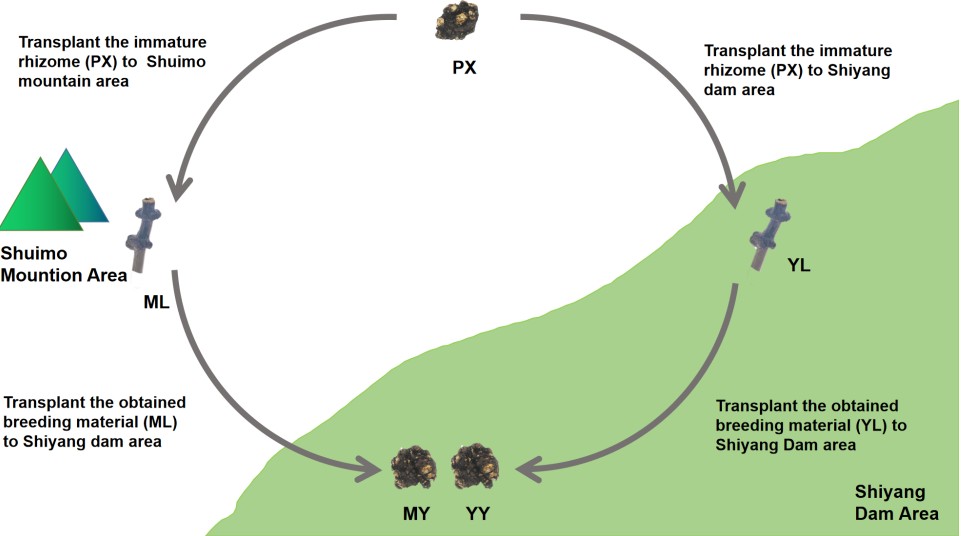

**Figure 2** **The design of cultivation experiment of *L. chuanxiong*.** ML, stem node obtained from mountain cultivation; MY, rhizome obtained by transplanting ML to dam area; YL, stem node obtained from dam cultivation; YY, rhizomes obtained by transplanting YL to dam area.

into sterile 50 mL centrifuge tubes, and finally stored in a refrigerator at −80 °C. (2) Plant sample

Plant samples were collected in the experimental field by random sampling method. After the surface was washed with sterile distilled water, the epidermis was scraped off with a sterile scalpel, and then sterilized with 2% sodium hypochlorite for 15 min. The samples were washed with sterile distilled water for 5 times, and stored in a refrigerator at −80 °C.

## DNA extraction and library construction

The sample was rapidly ground to a powder under liquid nitrogen. Take 0.5 g of sample powder, and use Zymo Research BIOMICS DNA Microprep Kit (Cat # D4301) to purify the sample gDNA with 3 biological repeats. The integrity of gDNA was detected using 0.8% agarose electrophoresis, followed by nucleic acid concentration detection using Tecan F200 (PicoGreen dye method). The 16S rDNA V4 region was amplified using Primer pair 515F (′5-GTGYCAGCMGCCCGCGGTAA-3′) and 806R (′5-GGACTACHVGGGTWTCTAAT-3′). GSS nucleic acid locking technology of Rhonin Biosciences Company was used to avoid interference of plant DNA. The procedure was as follows: pre denaturation 94 °C for 1 min, 1 cycle; Denatured at 94 °C for 20 s, annealed at 54 °C for 30 s and extended at 72 °C for 30 s, 25–30 cycles; 72 °C for 5 min, 1 cycle; 4 °C thermal insulations. Each sample was repeated for three times by PCR technology, and the PCR products in the linear phase were mixed in equal amounts. The PCR product was mixed with six-fold loading buffer, and then the target fragment was detected by electrophoresis using 2% agarose gel. The qualified amplification products were packed in dry ice and sent to Chengdu Rhonin Biosciences Co.,Ltd. for DNA high-throughput sequencing. PE250 sequencing method was adopted, and the Hiseq Rapid SBS Kit v2 (FC-402-4023 500 Cycle) kit of Illmina Company was used for high-throughput sequencing. Use NEBNext Ultra II DNA Library Prep Kit for Illuminata (NEB # E7645L) of NEW ENGLAND BioLabs to build the library. The data was uploaded to the cloud platform of Shanghai Majorbio Biomedical Technology Co., Ltd. for diversity analysis. The original data has been uploaded to NCBI: https://www.ncbi.nlm.nih.gov/bioproject/PRJNA937038.

## Statistical analysis

QIIME (v1.9.1; http://qiime.org/) was used to analyze sequences. After removing low quality or ambiguous sequences, use flash (v1.2.11; http://ccb.jhu.edu/software/FLASH/index.shtml) to splice the sequences. The Uparse (v11; https://drive5.com/uparse/) algorithm was used to perform OTU clustering on 97% similarity pair sequences, removing chimeras during the clustering process, and obtaining representative OTU sequences. RDP classifier (v2.13; https://john-quensen.com/classifying/rdp-classifier-updated/) was used to conduct taxonomy analysis on 97% of OTU representative sequences at similar levels, and the community composition of each sample was counted at each taxonomy level. The comparison database was the Silva138 bacteria database. In this study, we used mothur (v1.30.2; https://mothur.org/) to calculate alpha diversity index, including Shannon index and Ace index. The Student's $t$-test was chosen to test for inter group differences. All other analyses were conducted in R (v3.3.1; R Core Team, 2014).
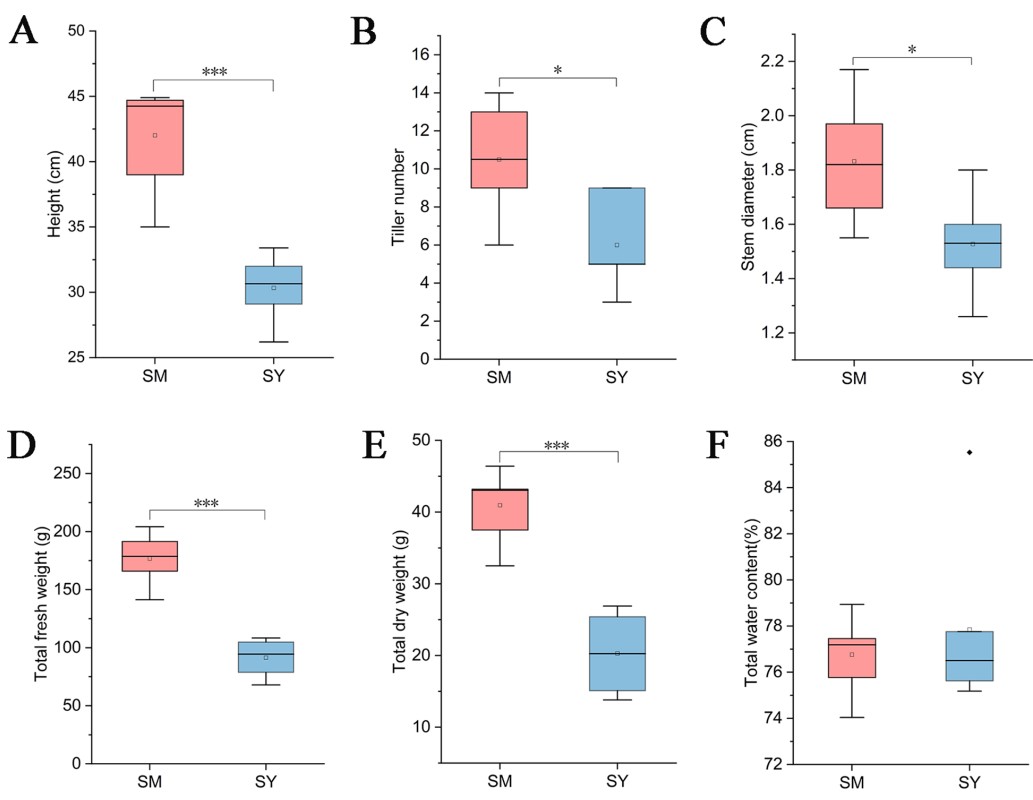

**Figure 3** **Analysis of growth index of *L. chuanxiong* in reproductive stage.** SM represents *L. chuanxiong* during the mountain breeding stage, and SY represents *L. chuanxiong* during the dam breeding stage. (A–F) The significant differences in plant height, tiller number, stem diameter, total fresh weight, total dry weight, and total water content between the two stages of *L. chuanxiong* (Students, *t*-test, $p < 0.05$).

## RESULTS

### Growth index of *L. chuanxiong* cultivated in mountainous and dam area

While collecting LZ samples, we measured the growth index of *L. chuanxiong*. The results showed that the plant height, tiller number, stem diameter, total fresh weight and total dry weight of *L. chuanxiong* were significantly higher than those of *L. chuanxiong* cultivated in Dam area after cultivation in mountainous area (Figs. 3A–3E), but the water content of *L. chuanxiong* was not significantly different (Fig. 3F, Students, *t*-test, $P < 0.05$).

### Diversity of bacterial communities

We analyzed the 16srNA sequencing results of 51 samples. We obtained a total of 1,089,819 reads after quality filtering, with a single sample sequence average of 21369. We performed OTU clustering on non repeating sequences based on 97% similarity, and removed chimeras during the clustering process, and ultimately obtained 5368 OTUs.

The OTU data set of each group sample was analyzed by the index inter group difference test. We found that after transplanting PX to the mountain or dam area, the richness of endophytic bacteria in harvested LZ showed significant changes ($P \leq 0.001$). The species

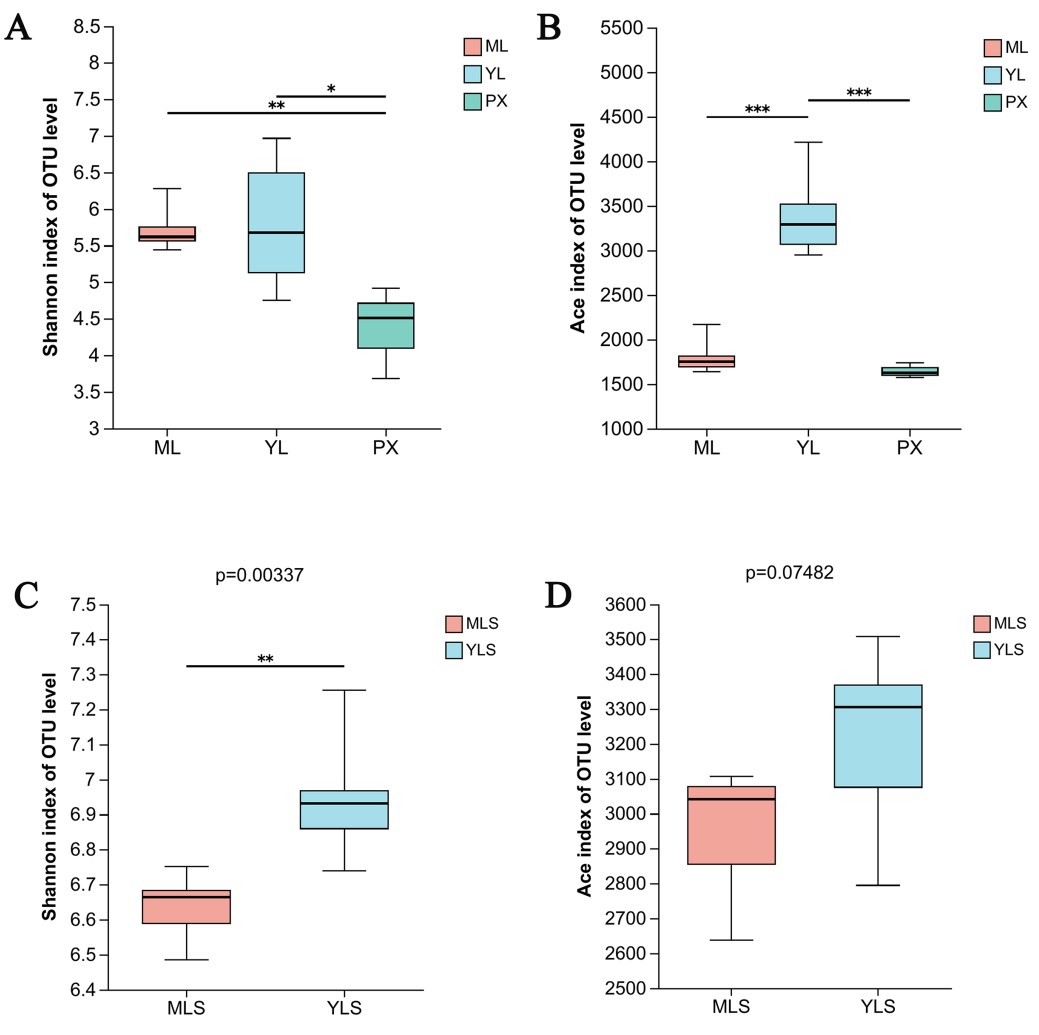

**Figure 4   Species diversity (shannon) and richness (Ace) of LZ and soil bacteria in mountainous and dam areas.** (A & B) Shannon index and ace index of ML, YL and PX, respectively; (C & D) Shannon index and ace index of cultivated ML soil (MLS) and cultivated YL soil (YLS), respectively.

diversity of ML and YL was higher than that of PX (Fig. 4A). At the same time, the richness of YL was significantly higher than that of ML and PX (Fig. 4B), indicating that the species diversity of endophytic bacteria increases after PX transplantation. From the soil analysis results, the bacterial communities of MLS and YLS had significant differences in species diversity, and the species diversity of YLS was significantly higher than that of MLS (Figs. 4C and 4D), indicating that the dam area soil and its cultivated LZ had greater bacterial diversity than the mountain soil and its cultivated LZ.

From the bacteria species Venn diagram of the sample, we could see that 1,479 OUT exist stably in the growth and development of *L. chuanxiong* under different reproduction modes, and were not affected by transplanting and planting stages. However, the transplanting in different regions also led to the difference of LZ bacterial community (Fig. 5A). Comparing LZ with the corresponding soil, we could find that 74.38% of

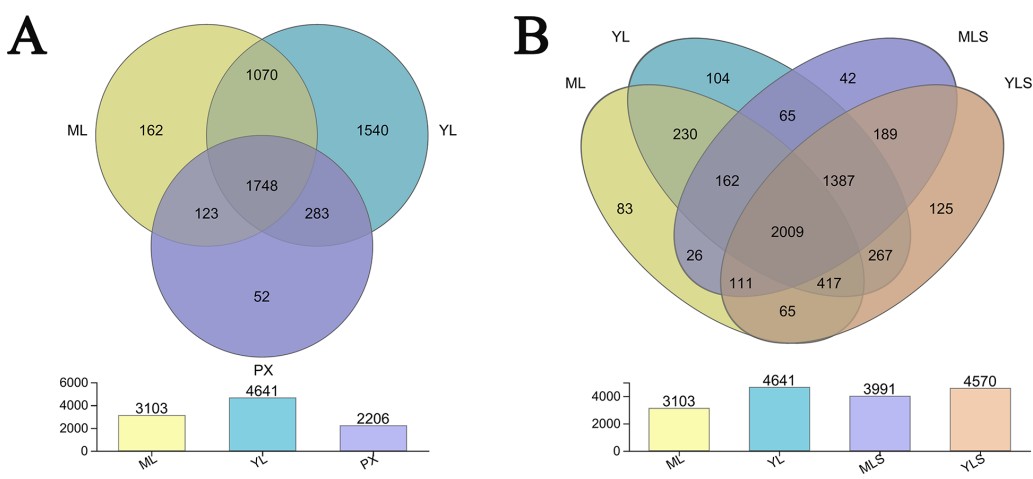

**Figure 5** **Distribution of bacterial OTU number in *L. chuanxiong* and its corresponding soil.** (A) Distribution of OTU numbers in ML, YL, and PX. (B) Distribution of OTU number in ML, YL, MLS, and YLS.

bacteria in ML are found in MLS, and 87.91% of bacteria in YL are found in YLS (Fig. 5B). From the above analysis, it can be seen that although some bacteria are stable in the process of transplanting and breeding, the transplanting of host plants to different regions still affected the composition of bacterial communities in LZ, and the transformation of endophytic bacterial community structure in LZ caused by transplanting to dam area was more than to mountain area.

## The bacterial community structures

Through hierarchical cluster analysis (Fig. 6A), we could see that MLS and YLS bacterial communities were separated. The bacterial communities of MYS and YYS did not gather together, too. It could be seen that the bacterial communities of soils (MLS and YLS) in the mountain region at Shuimo and the Dam region at Shiyang were different, and the bacterial communities of *L. chuanxiong* rhizosphere soils of the same Dam reagion at Shiyang (MYS, YYS) were also different while its LZ came from different regions (ML which cultivated from mountain region at Shuimo and YL which cultivated from the Dam region at Shiyang). The clustering results of plant samples (Fig. 6B) show that ML and YL were separated, but MY and YY are gathered together. It was obvious that the structure of LZ endophytic bacteria cultivated in different soil environments had changed differently. However, after the two types of LZ are transplanted to the same dam area and continue to be planted, the host endophytic bacteria tend to recombine.

The principal coordinate analysis (PCoA) based on the Bray Curtis matrix algorithm could clearly reflect the difference between the endophytic bacterial community of *L. chuanxiong* and the bacterial community of soils. The first principal component respectively explained 13.35% of the total variation of bacterial community in *L. chuanxiong* (Fig. 7A) and 26.96% of the total variation of bacterial community in soil (Fig. 7B). At the same time, the bacterial communities of ML and YL were separated, and the bacterial communities of

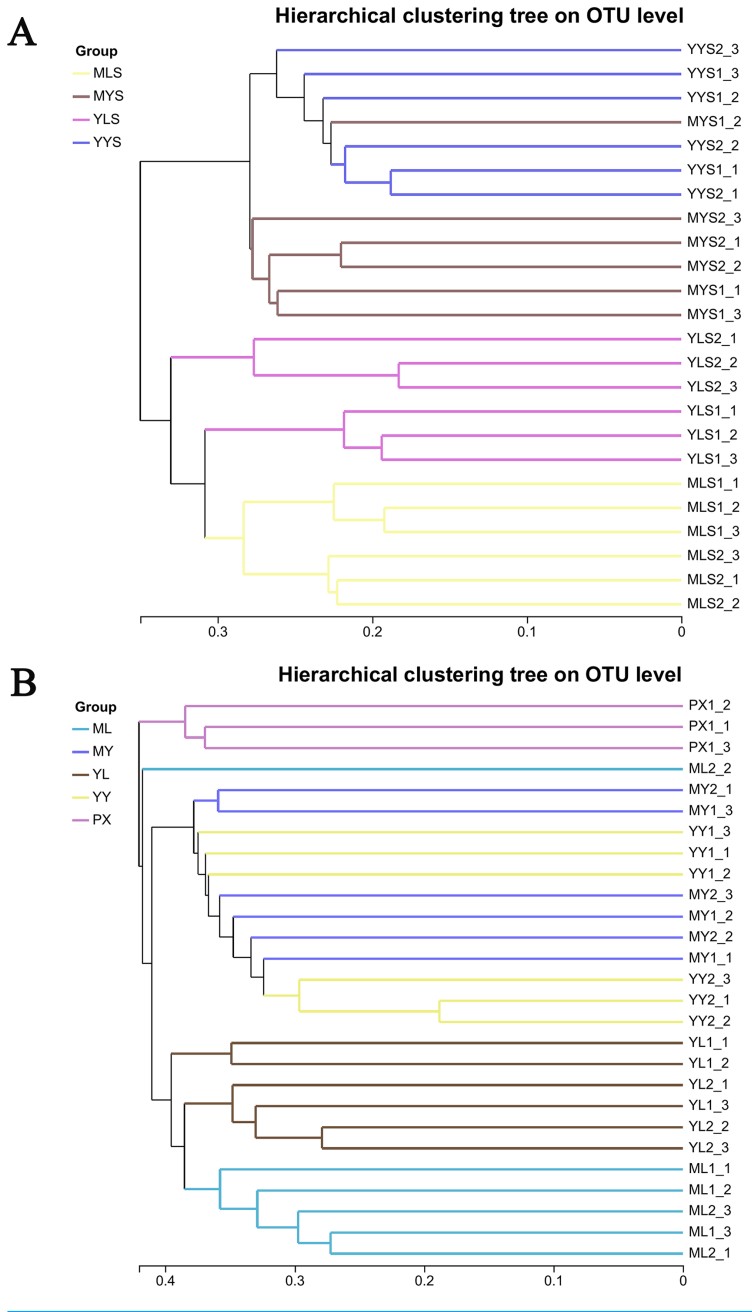

**Figure 6** **Hierarchical cluster analysis of *L. chuanxiong* and soil samples.** (A) The hierarchical clustering analysis of plant samples PX, ML, MY, YL, YY. (B) The hierarchical clustering analysis of soil samples MLS, MYS, YLS and YYS.

MLS and YLS were also separated, which was similar to the results of hierarchical cluster analysis.

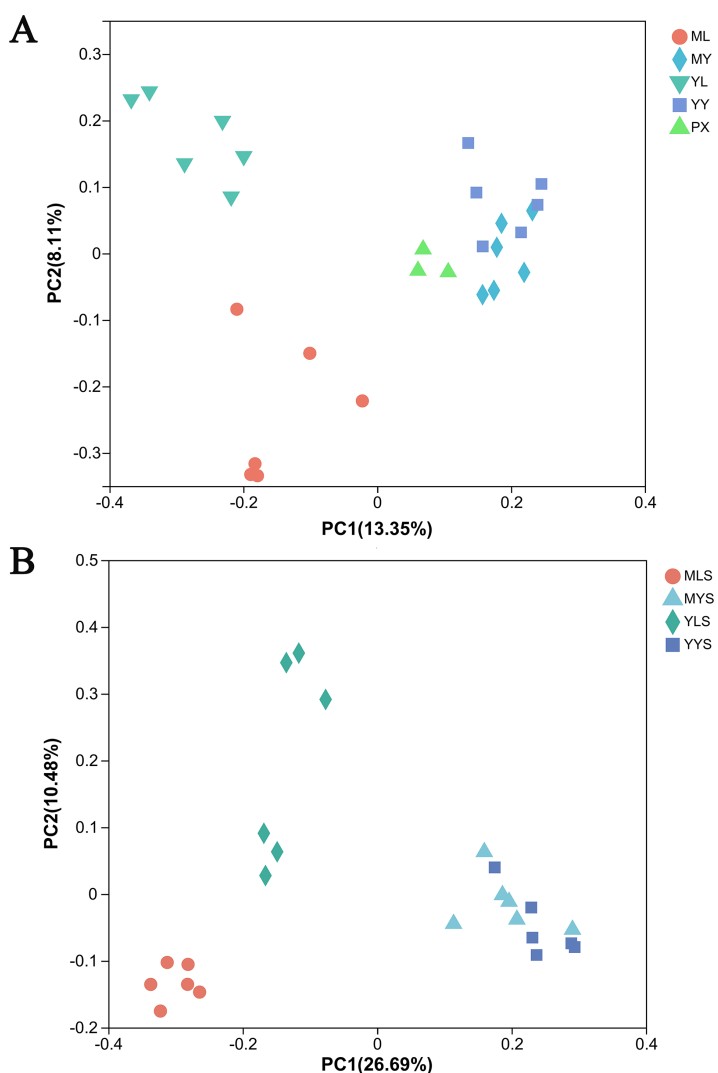

**Figure 7** **PcoA analysis of *L. chuanxiong* and soil sample.** (A) The PcoA analysis of plant samples PX, ML, MY, YL, YY. (B) The PcoA analysis of soil samples MLS, MYS, and YYS.

## Analysis of bacterial community composition

We analyzed the community composition of endophytic bacteria in the plant and soil samples of *L. chuanxiong* under two transplanting modes. At the level of phyla classification, it could be seen from the analysis histogram of the plants sample community composition (Fig. 8A) that the relative abundance of Proteobateria, Bacteroidota, Actinobacteriota, and Acidobacteriota were significantly higher than that of others, and the relative abundance content of each of them was more than 1%. Inside, the relative abundance of Proteobacteria in the sample was more than 50%, which shows that it occupies the most important niche in *L. chuanxiong*. In addition, Actinobacteriota are also noteworthy group. Through the difference test between the two sample groups (Figs. 8C and 8D), it can be seen that the relative abundance of Actinobacteriota in ML is significantly higher than that in YL

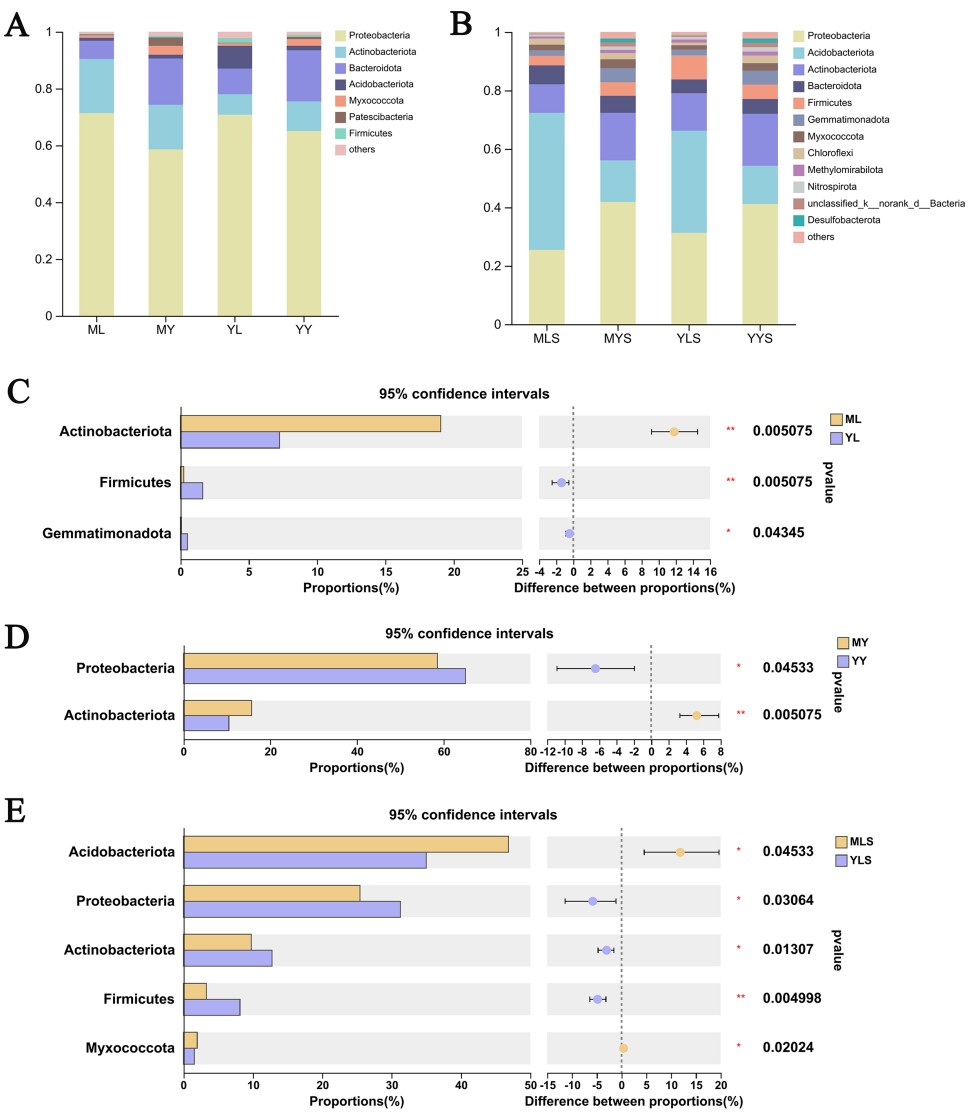

**Figure 8** **Analysis chart and difference of bacterial community composition at phyla level.** (A) Community composition of plant samples ML, MY, YL, YY at the phyla level; (B) community composition of soil sample MLS, MYS, YLS, YYS at the phyla level; (C) community composition difference of ML and YL at phyla level; (D) community composition difference of MY and YY at phyla level; (E) community composition difference between MLS and YLS at phyla level.

(Wilcoxon rank_sum test, $P < 0.05$), while the relative abundance of Actinobacteriota in MY is also significantly higher than that in YY, with a significant difference (Wilcoxon rank_sum test, $P < 0.05$), indicating that the dominant flora of mountain breeding cluster continues to exist in the rhizome.

The analysis of soil community composition (Fig. 8B) showed that the abundance of Proteobateria, Acidobacteriota, Actinobacteriota, Bacteroidota, and Firmicutes was significantly higher than that of other bacteria, greater than 1%, which was similar to that of plant samples. Comparing the soil (Fig. 8E), we found that Acidobacteriota in MLS is

significantly higher than that in YLS, but Proteobacteria, Actinobacteriota was significantly lower than YLS, which was inconsistent with the difference rule between ML and YL. At the same time, we also found that plant sample and soil had different preferences for bacterial community. The relative abundance of Proteobacteria in plant sample was significantly higher than that in soil, while the relative abundance of Acidobacteriota in soil is significantly higher than that in plant sample. The above results indicated that, after transplanting to different environments for reproduction, LZ had constructed different endophytic bacterial communities, but the bacterial communities enriched in the host plants might not be the bacterial communities with absolute superiority in the soil (Wilcoxon rank sum test, $P < 0.05$).

At the family level (Fig. 9A), we could see that the abundance of 12 families of plant sample was higher, all greater than 1%. According to the analysis of the difference between groups of LZ from two places (Fig. 9C), the relative abundance of Xanthobacteraceae, Micromonosporaceae, Beijerinkiaceae, Rhodanobacteria, Sphingomonadaceae, Microbacteria, Nocardioidaceae, Streptomycetaceae in ML were significantly higher than that in YL (Wilcoxon rank sum test, $P < 0.05$), and this part of endophytic bacteria is mainly composed of Proteobateria and Actinobacteriota. However, the relative abundance of norank_o__ Vicinamibacteralea, Vicinamibacteraceae, Flavobacteriaceae and Pectobacteriaceae in YL was significantly higher than that in ML. The endophytic bacteria community in rhizome of *L. chuanxiong* (Fig. 9D) showed that that the relative abundance of Rhodanobacteria, Micromonosporaceae, Chitnophagaceae and Streptomycetaceae in MY were significantly higher than that in YY, while the relative abundance of Xanthomonadaceae was significantly lower than that in YY, of which Micromonosporaceae and Streptomycetaceae belonged to Actinobacteriota. It could be seen from the analysis of soil bacterial community composition (Fig. 9B) that the relative abundance of 14 families in the soil reached 1%, which was similar to the composition of bacterial communities in plant samples. Compared with the soil (Fig. 9E), we found that the relative abundance of norank-o-Vicinamibactrales, Chitnophagaceae and Bryobactaceae in MLS were higher than YLS, with significant difference (Wilcoxon rank sum test, $P < 0.05$).

## DISCUSSION

### Transplantation led to the recombination of endophytic bacteria in LZ

In this study, we found that after transplanting, the bacterial diversity of LZ increased, and the species richness of endophytic bacteria in YL was significantly higher than that in ML. The bacterial communities of the two kinds of bulk soils were significantly different, and the composition of LZ bacterial communities was similar to that of the bulk soils. A total of 74.38% of ML bacterial community was consistent with MLS, and 87.91% of YL bacterial community was consistent with YLS. These results showed that, after transplanting, LZ endophytic bacteria recombined. The degree of recombination of endophytic bacteria community caused by local transplanting was greater than that caused by mountain transplanting, which was just opposite to the recombination of fungal community of *L. chuanxiong* in the previous study (*Kang et al., 2021*). This might be related to the species

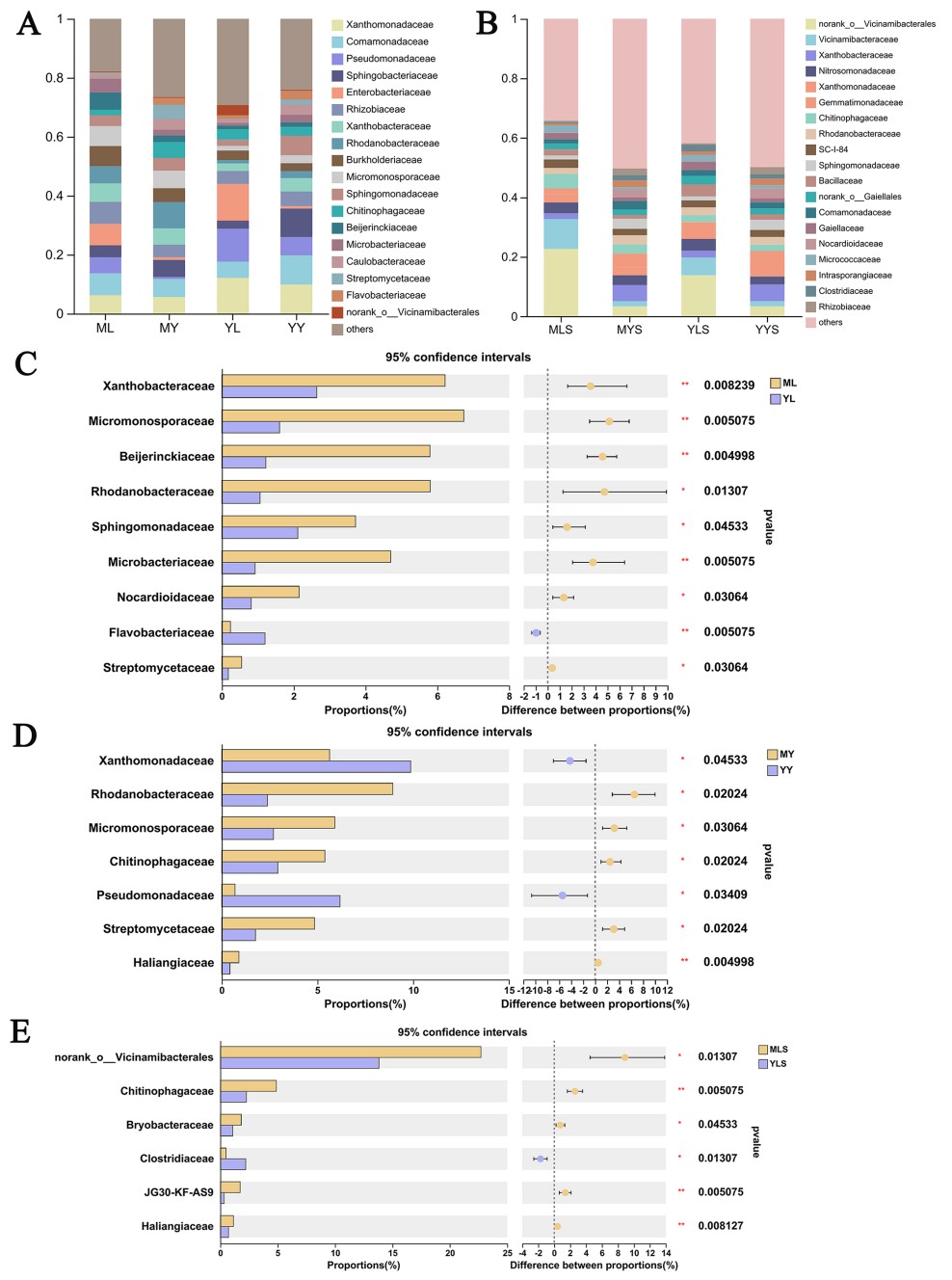

**Figure 9  Analysis chart and difference of bacterial community composition at family level.** (A) Community composition of plant samples ML, MY, YL, YY at the family level; (B) community composition of soil sample MLS, MYS, YLS, YYS at the family level; (C) community composition difference of ML and YL at family level; (D) community composition difference of MY and YY at family level; (E) community composition difference between MLS and YLS at family level.

diversity of the bacterial community in the soil: the species diversity in the mountain soil was lower than that in the dam soil. The difference of bacterial community in the soil may be related to the soil type and the physical and chemical properties of the soil (*Islam et*

*al., 2020*), which needs to be further verified by subsequent experiments. No matter what caused the difference of bacterial community structure in soil of mountain and dam area, transplanting provided an important opportunity for the recombination of endophytic bacterial community of host plants.

In this study, we found that the relative abundance of Xanthobacteraceae, Beijerinkiaceae, Rhodanobacteraceae, and Sphingomonadaceae in Proteobateria in ML was significantly higher than that of YL, among which Xanthobacteraceae and Sphingomonadaceae were reported to be electrochemical active bacteria (*Zhang et al., 2018*), while Rhodanobacteraceae was a typical denitrifying-bacteria (*Liang et al., 2021*), both of which had a decomposition effect on nitrate in soil (*Zhu et al., 2021*). These bacteria are enriched in the stem nodes of *L. chuanxiong* in mountainous areas, which may promote the utilization of nitrate by plants, such as promoting photosynthesis and improving stress resistance (*Guo et al., 2019*; *Hai An et al., 2018*). At the same time, the tiller number and stem diameter of *L. chuanxiong* in mountainous areas were significantly higher than those in dam areas, indicating that these bacteria may be involved in the absorption and utilization of nutrients in the stem nodes of *L. chuanxiong*, thereby promoting its growth. Beijerinkiaceae bacteria are reported to be more abundant in areas with more greenery, lower air pollution, and lower temperatures (*Babette et al., 2022*), suggesting that mountainous climate conditions may be an important reason for the enrichment of certain bacteria.

When analyzing soils, we found that the relative abundance of Acidobacteriota was significantly higher in MLS than in YLS. Acidobacteriota is one of the most dominant and abundant bacteria in soil (*Challacombe & Kuske, 2012*), and its diversity and abundance of bacterial species, as well as its diversity of metabolic traits, make it a potentially important bacterium in soil nutrient cycling (*Kielak et al., 2016*). It has been reported that Acidobacteria in soil may be an important component of various ecosystems and a driving factor of different ecosystem processes (*Kielak, Van Veen & Kowalchuk, 2010*). Zhang studied the relative abundance of Acidobacteria at different altitudes in Shennongjia, and found that the altitude has a significant impact on the abundance of Acidobacteria. With the increase of altitude, the relative abundance of Acidobacteria also increases. He also found that most of the bacteria in Acidobacteria can decompose starch in the soil (*Zhang et al., 2014*), which may help plants to use carbon sources in the soil. Under the two breeding modes, the planting soil elevation of LZ is different, which may be the reason for the difference in the relative abundance of Acidobacteria in the soil of the two places. In addition, the relative abundance of Chitinophagaee in MLS at the family level is also higher than YLS. It has been reported that three new species of Chitinophagaee were isolated from cowpea rhizosphere soil, which can produce indole-3-acetic acid and NH3, and have the ability to dissolve phosphate, showing the role of promoting plant growth (*Munusamy et al., 2015*). Our previous research had separated and identified some dominant endophytic fungi from ML, which could produce auxin and gibberellin, and promote the growth and development of ML. At the same time, it promoted the formation of LZ and increased the output of asexual propagation materials. The results of growth indicators showed that the mountain environment played a role in promoting the growth of *L. chuanxiong*. Those

bacteria with potential growth promotion function might also be the key factors in the growth process of *L. chuanxiong*. Whether the dominant bacteria in ML found in this study also have similar functions needs further experiments to explain.

Reciprocal relationships in the rhizosphere are illustrated by the fact that plants can select soil microorganisms to reshape the endophytic microflora (*Berendsen et al., 2013*). The process of transplanting led to significant differences in endophytic bacterial communities in both upland and flat dam propagation, indicating that the bacterial community in soil had an impact on the assembly of endophytic bacterial communities in plants (*Garbeva, Van Veen & Van Elsas, 2004*). Different soil microecological environment may be the cause of recombination, which is consistent with the research of *D'alessandro et al. (2014)* and *Emami et al. (2019)*. *Afzal et al. (2019)* also believes that the diversity of endophytic bacteria depends on specific factors such as various bacteria, plants and environment. The dominant bacterial community in mountain environment and in ML might be one of the reasons for the difference of breeding material germplasm. Up to now, there have been few reports about transplanting propagation on asexual reproduction in the previous literature; the deeper impact of transplanting on the recombination of endophytic bacteria in asexual reproduction plants needs further research.

## More Actinobacteriota with potential disease resistance and growth promotion function were enriched in ML

During plant growth, the relative abundance of some bacterial communities increases while others decrease, and the abundance of bacterial communities in plants is dynamic (*De Araujo et al., 2019*). In the cultivation stage of *L. chuanxiong* (PX-LZ-CX), regardless of the breeding mode, the abundance of Proteobateria continues to decrease, while the relative abundance of Bacteroidota decreases first and then increases, indicating that the change trend of some endophytic bacterial communities is consistent and is not affected by the breeding mode. However, some endophytic bacterial communities have undergone different changes under the influence of transplanting patterns. Among them, Actinobacteriota are the most obvious. At the phylum level, the relative abundance of Actinobacteriota in ML was significantly higher than that in YL, and this change continued in MY and YY. Further analysis also found that the relative abundance of Micromonosporaceae, Microbacteriaceae, Nocardioidaceae, Streptomycoticaceae under the Actinobacteriota gate in ML was also significantly higher than that in YL. As we all know, Actinobacteria is an extremely precious treasure house of microbial resources, contributing 45% of the bioactive substances (*János, 2005*) in microbial resources, and has great prospects for development and application. It inhibits plant diseases by producing antibiotics, and promotes plant growth by dissolving phosphate and generating secondary metabolites (*Franco-Correa et al., 2010*; *Palaniyandi et al., 2013*). Micromonosporaceae is widely distributed in nature, including terrestrial, aquatic and plant endophytic environments (*Zhang, Xin & Zhao, 2019*; *Ramesh & William, 2013*; *Nattakorn et al., 2018*). Micromonosporaceae can produce aminoglycosides or alkaloids and other antibiotics (*Armstrong et al., 2012*; *Charan et al., 2004*), which can inhibit gram-positive bacteria or viruses (*Wagman, 1980*). *Triningsih et al. (2022)* found Micromonosporaceae in the sea can

produce Paulomycin G, which is a new natural product with cytotoxicity to tumor cell lines (*Sarmiento-Vizcaíno et al., 2017*). *Jurado et al. (2019)* isolated 220 strains of Actinomycetes from soil compost, among which two strains under Microbacteriaceae were the best strains producing salicylic acid and chitin. These bioactive substances are indicators to describe the ability of strains in disease resistance and growth promotion. Streptomycetaceae is the most widely studied part of Actinobacteria, in which Streptomyces contributes about 80% of the total secondary metabolites of actinomycetes and is the main source of various antibiotics (*Watve et al., 2001*). Streptomyces isolated from different sources have many beneficial properties, such as plant pathogen inhibition (*Franco et al., 2007*), organic decomposition and plant growth promotion (*Passari et al., 2016*). Cycloheximide and streptomycin produced by Streptomyces griseus were the first antibiotics used to control plant fungal and bacterial diseases (*Leben & Keitt, 1954*). Soybean (*Nassar, El-Tarabily & Sivasithamparam, 2003*), tomato (*El-Tarabily, 2008*), wheat (*Sadeghi et al., 2012*) and other crops can promote plant growth by producing IAA to promote seed germination and root elongation. In addition, Streptomyces can also induce host plants' defense pathways to respond more quickly to pathogen attacks (*Conn, Walker & Franco, 2008*). The more abundant Actinobacteria in ML may indicate that the asexually propagated *L. chuanxiong* has enriched more bacteria beneficial to LZ, a propagation material, in the mountains, which might be the reason for the difference in resistance between ML and YL germplasm. Moreover, the phenomenon that more actinomycetes were enriched in LZ transplanted in mountainous area persisted in the rhizomes of subsequent cultivation.

In the early stage, the research group isolated and identified four actinomycetes (*He et al., 2016*) of Streptomyces that had strong inhibitory effects on gram-negative bacteria and fungi from healthy LZ of *L. chuanxiong*, and had obvious antagonistic effects on *Fusarium oxysporum* and *Fusarium solanacearum*, the main root rot pathogens of *L. chuanxiong*. Our study found that the abundance of actinomycetes in LZ of *L. chuanxiong* still existed in the medicinal rhizome, which was conducive to the maintenance of the resistance of *L. chuanxiong*. However, this resistance is time-sensitive. If it is propagated continuously in Dam area, it will lead to the degradation of LZ germplasm and the aggravation of disease, which needs to be returned to the mountain for reproduction. In this study, the results of high-throughput sequencing confirmed that ML has more abundant actinomycetes than YL, and MY also has more actinomycetes than YY, which further indicating that the microecological recombination of MY driven by transplantation may be beneficial to increase its disease resistance, so as to avoid serious diseases and insect pests caused by long-term asexual reproduction. The example of this kind of off-site transplanting that saved *L. Chuanxiong* from the crisis of germplasm degradation is worthy of being used for reference by other asexually propagated crops, such as *Angelica sinensis*.

## CONCLUSIONS

This study provided relatively comprehensive evidence for the influence of the microenvironment composed of endophytic bacteria from LZ obtained by different transplanting methods on the growth of LZ. Our results showed that the community

structure of endophytic bacteria of *L. chuanxiong* had changed in the process of planting, whether it was transplanted to the mountain or the dam area, indicating that the means of transplanting had an important impact on the recombination of endophytic bacteria. Soil environments were the main factor for the recombination of endophytic bacteria in *L. chuanxiong*. The mountainous soil environment made LZ obtain more endophytic bacteria with potential growth-promoting and biological control functions. Although local transplanting can also cause the endophytic bacteria recombined in LZ, it fails to enrich more actinomycetes and produce new endophytic bacteria community structure as in mountain breeding. It was not clear to what extent these bacteria affected the growth and health of *L. chuanxiong*. In the future, we can attempt to artificially construct cultivable synthetic communities to further explore the underlying mechanisms of these potentially beneficial endophytic bacteria on the growth of *L. chuanxiong*. This phenomenon provides us with an idea, that is, whether we can reconstruct the micro-ecological structure of the host, such as off-site transplantation to alleviate the problems caused by asexual reproduction.

## ACKNOWLEDGEMENTS

We are very grateful to Lei Kang and Hongyang Lv for their collection of Chuanxiong and soil samples, as well as the analysis platform provided by Shanghai Majorio Bio-pharm Technology Co., Ltd.

### Funding

This work was supported by the Ability Establishment of Sustainable Use for Valuable Chinese Medicine Resources (2060302-1702-01), the National Natural Science Foundation of China (81673553 and 81001610), and the Science and Technology Project of Sichuan Provincial (2021YJ0113). The funders had no role in study design, data collection and analysis, decision to publish, or preparation of the manuscript.

### Grant Disclosures

The following grant information was disclosed by the authors:
Ability Establishment of Sustainable Use for Valuable Chinese Medicine Resources: 2060302-1702-01.
National Natural Science Foundation of China: 81673553, 81001610.
Science and Technology Project of Sichuan Provincial: 2021YJ0113.

### Competing Interests

The authors declare there are no competing interests.

### Author Contributions

- Wanting Xiao performed the experiments, analyzed the data, prepared figures and/or tables, authored or reviewed drafts of the article, and approved the final draft.

- Zhanling Zhang performed the experiments, prepared figures and/or tables, and approved the final draft.
- Hai Wang conceived and designed the experiments, performed the experiments, authored or reviewed drafts of the article, and approved the final draft.
- Guiqi Han performed the experiments, prepared figures and/or tables, and approved the final draft.
- Zhu-Yun Yan conceived and designed the experiments, authored or reviewed drafts of the article, and approved the final draft.
- Dongmei He conceived and designed the experiments, authored or reviewed drafts of the article, and approved the final draft.

### Data Availability

The sequences are available at NCBI GEO: PRJNA937038.

### Supplemental Information

Supplemental information for this article can be found online at http://dx.doi.org/10.7717/peerj.15579#supplemental-information.

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
