# Peer review of "Recombination of endophytic bacteria in asexual plant Ligusticum chuanxiong Hort. caused by transplanting"

_PeerJ, doi:10.7717/peerj.15579_

## Round 0.1 · original submission · Major Revisions

In addition to reviewers comments, there are some areas that could be improved:

The manuscript could benefit from a more detailed discussion of the potential implications of the findings. While the authors mention the potential for these endophytic bacteria to affect plant growth and health, they do not go into much detail on how this might happen or what the implications might be.

The authors state that "further experiments are needed to reveal the mechanism" of how these endophytic bacteria affect the growth and health of L. chuanxiong. It would be helpful if the authors could provide some suggestions for future research directions to investigate these mechanisms.

The manuscript could benefit from more detailed information on the specific methods used to analyze the endophytic bacterial community, such as the sequencing approach and bioinformatics pipelines used. This would help readers better understand how the data were generated and analyzed.

While the authors mention that the means of transplanting had an important impact on the recombination of endophytic bacteria, they do not provide much detail on how this might be happening. It would be helpful if the authors could provide some hypotheses or possible explanations for why this is the case.

The manuscript could also benefit from more detailed information on the environmental factors that may be driving the changes in the endophytic bacterial community. While the authors mention that soil environments were the main factor for the recombination of endophytic bacteria in L. chuanxiong, they do not provide much detail on what specific soil factors might be driving this. Providing more information on this could help readers better understand the mechanisms behind the observed changes in the endophytic bacterial community.

Reviewer 1 ·

Basic reporting

The abstract presents an interesting study that explores the effects of transplanting cultivation materials of Ligusticum chuanxiong on the bacterial communities within the plant and soil. The study seeks to answer whether bacteria have the same rejuvenating effects as endophytic fungi, which have been previously found to help avoid germplasm degradation in this plant species.

The methods used in the study appear to be appropriate, with high-throughput sequencing used to analyze bacterial communities in the plant and soil samples. The results are presented clearly, and the authors have provided detailed information on the bacterial communities found in the different samples.

However, the abstract could be improved by providing more specific information on the potential implications and significance of the findings. For example, the abstract could highlight any significant differences in bacterial communities between the different samples and discuss the potential implications for plant growth and health.

Overall, the study appears to be a valuable contribution to the understanding of the effects of long-term asexual reproduction on plant germplasm and the potential role of bacteria in rejuvenating plant populations.

However, there are linguistic and scientific errors which should be corrected

In the Introduction, plenty of scientific names are written non-italic.

Experimental design

Overall, the Materials and Methods section provides a clear and detailed description of the experimental design, sample collection, DNA extraction, library construction, and statistical analysis methods used in the study. However, there are a few areas where more information could be provided:

Sample size: The number of samples collected for each treatment group and the number of biological replicates used for DNA extraction and library construction are not specified. Providing this information would help readers evaluate the statistical power of the study and the reliability of the results.

Quality control: While the section mentions that gDNA integrity was assessed using agarose gel electrophoresis and nucleic acid concentration was measured using the PicoGreen dye method, it is unclear whether any additional quality control measures were taken to ensure the accuracy and reproducibility of the sequencing data. For example, it would be useful to know if any negative controls were included to check for contamination, and if any steps were taken to address batch effects or sequencing errors.

Data analysis: The section briefly describes the software and databases used for OTU annotation and diversity analysis, but does not provide any information on the specific parameters or cutoffs used for filtering, clustering, or normalization. Additionally, it is unclear how the statistical significance of the results was determined or whether any correction for multiple testing was applied.

Validity of the findings

Findings are valuable and constructive.

·

Basic reporting

At first, I would to mention the actual topic of the study – to compare the microbial community of L. chuanxiong before and after transplanting.
Bioinformatics approaches described in the manuscript can be improved. There are strong reasons to use amplicon sequence variants (ASVs) instead of OTUs for less noise and avoid spurious taxons (https://www.mdpi.com/2306-5354/9/4/146).
Callahan, McMurdie and Holmes points that “the improvements in reusability, reproducibility and comprehensiveness are sufficiently great that ASVs should replace OTUs as the standard unit of marker-gene analysis and reporting” Callahan, B., McMurdie, P. & Holmes, S. Exact sequence variants should replace operational taxonomic units in marker-gene data analysis. ISME J 11, 2639–2643 (2017). https://doi.org/10.1038/ismej.2017.119

Experimental design

Some details of the study must be clarified to better understanding and reproducibility:
L139: what the database (“silver”) used for analysis? How quality filtering of reads have been performed?
L152: why no public release of raw data (FASTQ) was performed?
L191: text and figure 7a don’t meet, 26, 96% ≠ 26,69 %, 13,35% ≠ 24%.
As said above, using OTUs approach produces a lot of spurious OTUs. L155 shows hundreds of OTUs, most of which, I propose, share less than 1%. Such classification is hard to interpret. For further study see, f.e.:
https://doi.org/10.1371/journal.pone.0227434
https://doi.org/10.1038/s43705-021-00033-z
• L312: what does mean “gate level”?
• L195: sentence better place to Discussion
• L318: references [10], [11], [17], [38] and many others has no journal specified;

Validity of the findings

L276: The role of Bacteroidetes, which are well-known digesters of organic matter, doesn’t considered
L207: sentence talks about “groups”, but one group mentioned only
My notes about the main text:
• L150: Fig. 3e -> 3f (water content)
• L152: missing space
• “(Fig. 9d)” repeating text
• L327: these abbreviations not used in further text
• L163: Fig. 4d no significance markers (p-value); Fig. 5a has no YL signature at the diagram

Reviewer 3 ·

Basic reporting

pass

Experimental design

pass

Validity of the findings

pass

Additional comments

I have reviewed the manuscript “Recombination of endophytic bacteria in asexual plant Ligusticum chuanxiong Hort. caused by transplanting". The authors studied how L. chuanxiong transplantation is affect by the bacterial community of endophytic bacteria. The content is interesting for the target audience of the journal and is generally well written.
However, the authors should add more details in the method section on how the 16s sequencing data is processed (L116 – L143). For example, how the QC of the sequencing data is done, which software is used to trim the sequencing data, which normalization method is used for the OTU data.
Apparently the health/growth of the plants is affected both by the endophytic bacteria and soil bacteria. The authors should try PCA analysis or variation analysis, or any other similar statistical analysis with both rhizosphere community and endophytic community to check whether there is a strong correlation between these two communities.

---

## Round 0.2 · Minor Revisions

The manuscript can be accepted after minor revisions.

·

Basic reporting

Some minor text corrections required:
L32-33: why "LZ" abbreviation is duplicated, not clear what LZ stands for; only in further text we see the meaning (L88)
The raw data reference (https://www.ncbi.nlm.nih.gov/bioproject/PRJNA937038) is still missing in the main text depite the answer in the rebuttal letter.

Experimental design

pass

Validity of the findings

pass

Reviewer 3 ·

Basic reporting

The authors have addressed the comments properly.

Experimental design

no comment

Validity of the findings

no comment

---

## Round 0.3 · accepted · Accept

The authors have revised the manuscript as per the suggestions of reviewers.